# Online searches of children's oseltamivir in public primary and specialized care: Detecting influenza outbreaks in Finland using dedicated databases for health care professionals

**Milla Mukka** [1]*, **Samuli Pesälä** [1,2], **Aapo Juutinen** [3], **Mikko J. Virtanen** [3], **Pekka Mustonen** [4], **Minna Kaila** [5], **Otto Helve** [3,6]

**1** University of Helsinki, Helsinki, Finland, **2** Epidemiological Operations Unit, City of Helsinki, Helsinki, Finland, **3** Department of Health Security, Finnish Institute for Health and Welfare, Helsinki, Finland, **4** Duodecim Medical Publications Ltd, Helsinki, Finland, **5** Clinicum, University of Helsinki, Helsinki, Finland, **6** Children's Hospital, Pediatric Research Center, University of Helsinki and Helsinki University Hospital, Helsinki, Finland

* milla.mukka@helsinki.fi

**Data Availability Statement:** All relevant data are within the manuscript and its Supporting information files.

## Abstract

### Introduction

Health care professionals working in primary and specialized care typically search for medical information from Internet sources. In Finland, Physician's Databases are online portals aimed at professionals seeking medical information. As dosage errors may occur when prescribing medication to children, professionals' need for reliable medical information has increased in public health care centers and hospitals. Influenza continues to be a public health threat, with young children at risk of developing severe illness and easily transmitting the virus. Oseltamivir is used to treat children with influenza. The objective of this study was to compare searches for children's oseltamivir and influenza diagnoses in primary and specialized care, and to determine if the searches could aid detection of influenza outbreaks.

### Methods

We compared searches in Physician's Databases for children's oral suspension of oseltamivir (6 mg/mL) for influenza diagnoses of children under 7 years and laboratory findings of influenza A and B from the National Infectious Disease Register. Searches and diagnoses were assessed in primary and specialized care across Finland by season from 2012–2016. The Moving Epidemic Method (MEM) calculated seasonal starts and ends, and paired differences in the mean compared two indicators. Correlation was tested to compare seasons.

### Results

We found that searches and diagnoses in primary and specialized care showed visually similar patterns annually. The MEM-calculated starting weeks in searches appeared mainly in the same week. Oseltamivir searches in primary care preceded diagnoses by −1.0 weeks

**Funding:** The author(s) received no specific funding for this work.

**Competing interests:** MK has held various trustee positions in the Finnish Medical Society Duodecim since the late 1990s. OH has held various trustee positions in the Finnish Medical Society Duodecim and Duodecim Medical Publications Ltd since 2009 and is a partner at iHealth Finland Ltd. The other authors have no competing interests. This does not alter our adherence to PLOS ONE policies on sharing data and materials.

**Abbreviations: GFT**, Google Flu Trends; **HCPs**, health care professionals; **ICD-10**, International Statistical Classification of Diseases and Related Health Problems, 10th Revision; **MEM**, Moving Epidemic Method; **NIDR**, National Infectious Disease Register; **PD**, Physician's Databases.

(95% CI: −3.0, −0.3; p = 0.132) with very high correlation ($\tau = 0.913$). Specialized care oseltamivir searches and diagnoses correlated moderately ($\tau = 0.667$).

## Conclusion

Health care professionals' searches for children's oseltamivir in online databases linked with the registers of children's influenza diagnoses in primary and specialized care. Therefore, database searches should be considered as supplementary information in disease surveillance when detecting influenza epidemics.

## Introduction

Influenza is a serious infectious disease appearing worldwide and is a significant public health concern. Following temporal patterns, influenza occurs in the cold seasons of the year in both adults and children, and infection is spread by two types of influenza viruses, A and B [1]. The global annual attack rate of influenza is 20–30% in children [2]. In a prospective cohort study on the incidence of influenza in Finnish children, an overall influenza attack rate was 19% (252/1,338) [3], and 15–20% of all respiratory infections tested as influenza-positive (3,637 specimens per season) during the epidemic peak [3]. Another Finnish study has shown that the burden of influenza is greatest in children under 3 years in terms of morbidity, complications, and treatment, as well as the significant parental work loss in caring for a sick child [4]. Antiviral agents, such as oseltamivir, are available to treat seasonal or pandemic influenza in children [5–7]. Oseltamivir is a neuraminidase inhibitor that prevents reproduction of the influenza virus and is available as a tablet or in a liquid form (oral suspension). The oral suspension of oseltamivir is typically used to treat children [7].

When using traditional data sources (e.g., diagnoses and laboratory findings) in infectious disease surveillance to detect outbreaks, reporting delays may exist. Online sources, such as general search engines, may provide unreliable information for epidemiological and surveilling purposes stemming from vague user profiles, poor search terms, and the impact of media coverage. Certain searches on Google Flu Trends (GFT) have coincided with medical visits related to influenza-like symptoms [8]; however, one study found that GFT data may not provide reliable surveillance for seasonal or pandemic influenza [9]. Nevertheless, searches have been used in estimating geographical influenza activity, thus making detection and surveillance of influenza epidemics possible [10, 11]. Combining information from several real-time flu predictors (e.g., hospital visits, Google Trends, Twitter posts, FluNearYou, GFT) has been shown to produce more accurate and robust real-time flu predictions [12]. Using machine-learning methods to combine these sources further improves influenza surveillance with more accurate and timely predictions [13, 14]. However, little data exist on searches of dedicated online databases used by health care professionals (HCPs) when detecting influenza outbreaks.

In Finland, a public health care sector comprises public primary and specialized care. Primary care includes health care centers where HCPs encounter first-point-of-contact patients with various symptoms and medical conditions [15, 16]. Specialized care focuses on patients with specific diseases, including medical specialists within their own specialties [15, 16]. Thus, primary care involves patients with unselected and undiagnosed symptoms, while specialized care diagnoses and treats patients with certain medical problems. Different working environments may reflect different information seeking needs in primary and specialized care [17–20]. HCPs seek medical information as part of their clinical work and increasingly use online

sources [19, 20]. Pediatricians in specialized care work in emergency departments, outpatient clinics, and hospital wards, handling children during influenza epidemics. Similarly, primary care physicians in health care centers also diagnose and treat children with influenza. In both sectors, physicians prescribe medication to children.

Children spread influenza easily during seasonal epidemics, and specifically those under 5 years of age risk developing severe influenza illness [21]. Oral suspension of oseltamivir is often prescribed in children [7]; however, errors in dosages may occur when calculated based on a child's weight or age [22, 23]. Therefore, information seeking to verify correct dosages highlights the value of real-time and reliable medical sources used in public health care centers and hospitals. Little is known about this information seeking behavior and its association with seasonal influenza outbreaks. Our previous study [24] found that HCPs' online searches of oseltamivir and influenza coincide with epidemiological data on influenza, thus we stated that searches could be used as an additional source of information for detecting influenza outbreaks. The aim of our current study is to compare HCPs' database searches for children's oseltamivir in public primary and specialized care, and to determine if they could be used as a supplementary source of information for detecting influenza epidemics. We hypothesized that HCPs' searches for oral suspension of oseltamivir in children would mimic the register data on children's influenza. This study provides novel information on dedicated online database use by HCPs in distinct health care sectors, and on whether information seeking behavior could predict influenza epidemics, as well as reveal the information needs of HCPs who diagnose and treat children with influenza.

## Materials and methods

### Setting

When seeking medical information online, HCPs in Finland use dedicated professional databases. Duodecim Medical Publications Ltd (owned by the Finnish Medical Society Duodecim) produces and maintains an online portal, Physician's Databases (PD) [25], allowing access to medical articles in Finnish. It is heavily used and available throughout the Finnish health care system, including public primary and specialized care. The number of opened articles can be tracked by an Internet protocol address. When searching for information on influenza, HCPs may access PD, which includes a pharmaceutical database with information on oral suspension of oseltamivir. Openings of the page with information on oral suspension of oseltamivir can be tracked in the log files of PD. In our study, we collected weekly online log data on the number of PD searches of oral suspension of oseltamivir (6 mg/mL) in primary and specialized care. In Finland, oral suspension of oseltamivir is the predominant antiviral medication for influenza in children under 7 years, and liquid form is rarely prescribed to any other patient group than children. Therefore, we considered oral suspension of oseltamivir as the most appropriate medication for studying influenza-related online searches related to children.

The Finnish Institute for Health and Welfare is the research and development institute in Finland that maintains the registers of public primary and specialized health care diagnoses and the National Infectious Disease Register (NIDR) [26]. NIDR includes positive laboratory findings of children's influenza A and B as notified electronically by microbiological laboratories. National registers of public primary and specialized care diagnoses collect information from electronic patient records about doctor visits when a physician reports a child's influenza diagnosis in the record. We compared the log data on oseltamivir to children's influenza diagnoses (J09–11 according to the *International Statistical Classification of Diseases and Related Health Problems*, 10th Revision [ICD-10] disease classification code system [27]) in primary

and specialized care, as well as positive laboratory findings of children's influenza A and B (under 7-year-olds) found from NIDR.

### Descriptive and statistical analysis

We analyzed the data across Finland from 2012 to 2016, comprising four influenza seasons (i.e., 2012/13, 2013/14, 2014/15, 2015/16) with six indicators (i.e., primary care searches of oral suspension of oseltamivir, specialized care searches of oral suspension of oseltamivir, children's primary care influenza diagnoses, children's specialized care influenza diagnoses, and children's positive laboratory findings of influenza A and B). The log data were analyzed anonymously, thus no individual HCP can be identified. No ethical approval was needed.

We used the Moving Epidemic Method (MEM) to calculate the length of each influenza season with starting and ending weeks and thresholds in four influenza seasons. MEM is a mathematical model assessing the timing of influenza epidemics based on historical data on weekly influenza rates [28]. The World Health Organization and the European Centre for Disease Prevention and Control have implemented MEM to monitor the circulation of influenza in European countries [28–30]. Paired differences in the starting weeks were calculated whether each indicator reaches the epidemic thresholds at similar times, comprising six indicators with a total of 15 pairs. As our data included a small number of searches, diagnoses, and laboratory findings (starting weeks), we used the bootstrapping method to estimate the distribution of observations [31]. The method bootstrapped paired differences consisting of six observations with 1,000 replications that yielded bootstrapped mean, bias-corrected mean, bias-corrected and accelerated (BCa) (adjusted for ties) 95% confidence interval (CI) of the mean, and p-value of the mean. A p-value <0.05 was considered statistically significant. We assessed season-to-season similarity between pairs by using Kendall's correlation coefficient ($\tau$). R was used to run MEM analyses (R version 3.6.3; packages "mem" and "boot") [32].

## Results

We found visually similar patterns in searches for oral suspension of oseltamivir and children's influenza diagnoses in primary and specialized health care in Finland during 2012–2016 by season (Fig 1). The number of searches in primary and specialized care remained similar (5,281 vs. 5,658), while the number of influenza diagnoses was only slightly higher (1.3x) in primary care than in specialized care (3,717 vs. 2,837). The number of laboratory reports of influenza A was almost three times higher (2.7x) than of influenza B (4,518 vs. 1,663). The number of searches, diagnoses, and laboratory reports of influenza is shown in Table 1.

The starts and ends of influenza seasons were calculated by using MEM. In primary care, searches for oseltamivir oral suspension started in weeks 51–4 and diagnoses in weeks 2–4 (Fig 2). In specialized care, oseltamivir searches started in weeks 50–4 and diagnoses in weeks 50–3 (Fig 3). In primary care, oseltamivir searches ended in weeks 8–14 and diagnoses in weeks 9–13. In specialized care, oseltamivir searches ended in weeks 9–14 and diagnoses in weeks 11–14. The season starts and ends of laboratory reports of influenza A appeared in weeks 50–3 and 9–14, while influenza B appeared in weeks 4–6 and 17–18. The pre-epidemic thresholds of oseltamivir searches in primary and specialized were 9 and 17, while diagnoses were 11 and 11. The post-epidemic thresholds of searches in primary and specialized care were 21 and 23, while diagnoses were 16 and 13. The MEM-calculated season starts and ends are shown in Figs 2 and 3 and Table 2.

Primary care searches for oral suspension of oseltamivir preceded children's influenza diagnoses by −1.0 weeks (95% BCa CI: −3.0, −0.3; p = 0.132) with very high correlation ($\tau$ = 0.913), while specialized care oseltamivir searches compared to diagnoses lagged by 0.3 weeks (95%

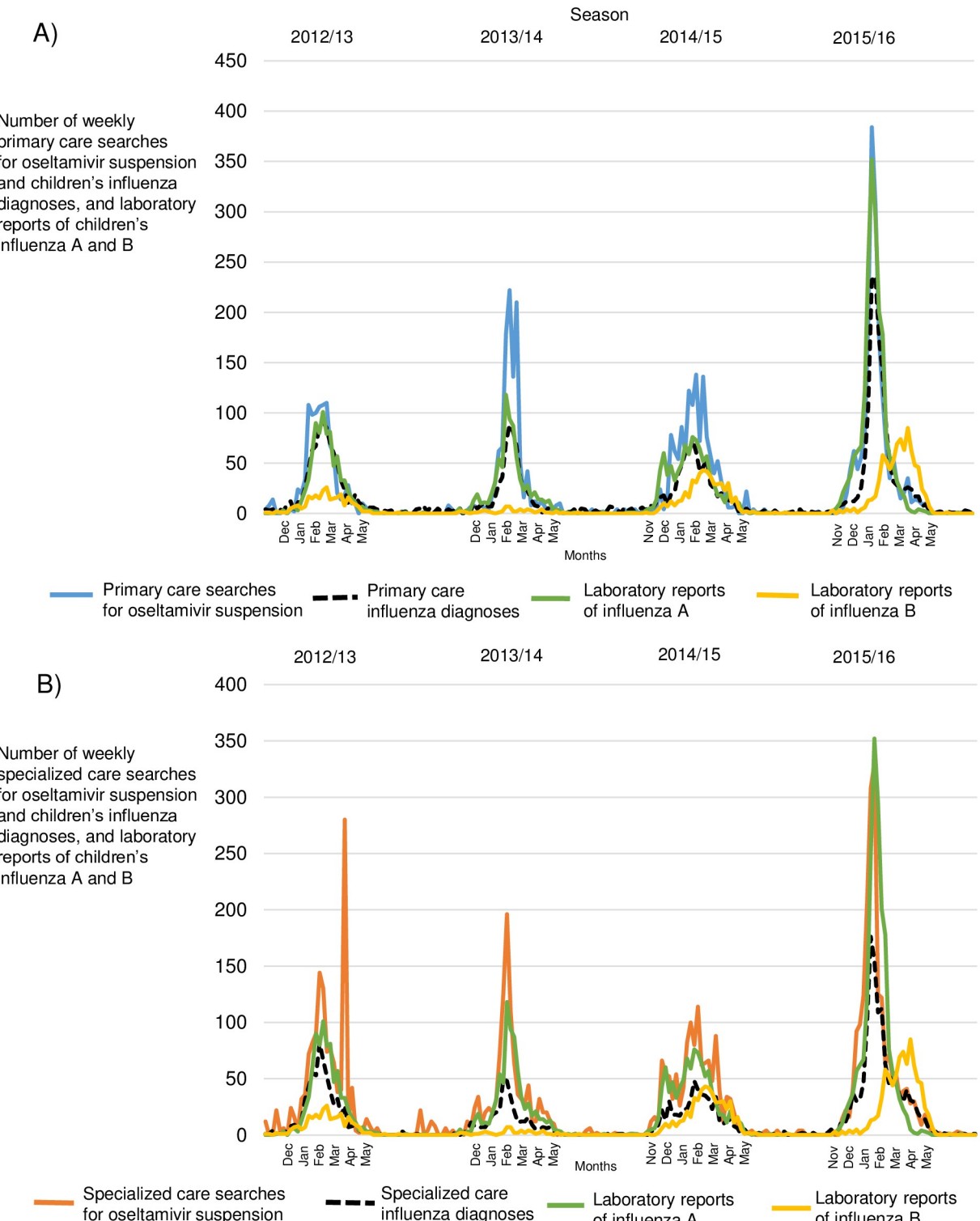

**Fig 1. Oseltamivir searches, influenza diagnoses, and laboratory reports of children's influenza A and B across Finland during 2012–2016 by season.** (A) Searches for oral suspension of oseltamivir and children's influenza diagnoses in primary care. (B) Searches for oral suspension of oseltamivir and children's influenza diagnoses in specialized care.

**Table 1. Total number of searches for oral suspension of oseltamivir, children's influenza diagnoses in primary and specialized care, and laboratory reports of children's influenza A and B across Finland 2012–2016 by season.**

| | Primary care | | Specialized care | | Laboratory reports | |
|---|---|---|---|---|---|---|
| | Number of searches for oral suspension of oseltamivir | Number of children's influenza diagnoses | Number of searches for oral suspension of oseltamivir | Number of children's influenza diagnoses | Number of children's influenza A | Number of children's influenza B |
| **Season** | | | | | | |
| **2012/13** | 996 | 867 | 1340 | 617 | 820 | 276 |
| **2013/14** | 1162 | 638 | 1080 | 373 | 773 | 62 |
| **2014/15** | 1314 | 798 | 1244 | 544 | 969 | 515 |
| **2015/16** | 1809 | 1414 | 1994 | 1303 | 1956 | 810 |
| **2012–16** | 5281 | 3717 | 5658 | 2837 | 4518 | 1663 |

BCa CI: −0.8, 1.3; p = 0.880) with moderate correlation (τ = 0.667). Laboratory reports of influenza A and specialized care influenza diagnoses showed very high correlation (τ = 0.913) with no gap in weeks (mean 0.0, 95% BCa CI: −1.0, 0.5; p = 0.001). However, laboratory reports of influenza A preceded primary care influenza diagnoses by −1.8 weeks (95% BCa CI: −3.5, −0.8; p = 0.006) with high correlation (τ = 0.800). In addition, very high correlations (τ = 0.913) were found between the following pairs: oseltamivir searches in specialized care and influenza diagnoses in primary care, laboratory reports of influenza A and influenza diagnoses in specialized care, and oseltamivir searches in specialized care and laboratory reports of influenza A. Weak or negligible correlations (0.000 < τ < 0.400) were found in pairs related to laboratory reports of influenza B. The pairs, paired differences, and correlations are shown in Table 3.

## Discussion

This study has shown that we could compare online searches for oral suspension of oseltamivir in primary and specialized health care to children's influenza diagnoses and laboratory reports of children's influenza A and B across Finland during 2012–2016 by season (Fig 1). Searches in primary and specialized care remained similar throughout seasons (Table 1), and starting weeks were calculated by using MEM (Figs 2 and 3). The results we found satisfied our hypothesis, suggesting that searches and diagnoses mimicked each other, mainly appearing in the same week (Table 2). Paired differences showed that searches preceded diagnoses only sometimes, and statistical significance was rarely found. Correlations were high in many pairs (Table 3), meaning that a paired indicator appeared similarly between seasons.

Prior research has shown that HCPs in specialized care use more online sources in their work compared to primary care. In hospitals, physicians and nurses favor online and other electronic sources of medical information [17, 19], while colleagues are the preferred information source among primary care physicians and nurses [20]. Interestingly, primary care physicians' information seeking from paper sources outnumbered online or electronic sources [18]. In our study, primary health care professionals performed a similar number of medical searches as those in specialized care (5,281 vs. 5,658), probably meaning that HCPs may prescribe oseltamivir similarly in both sectors. However, this needs further research. HCPs search for medical information during clinical work [19, 20], as well as during influenza epidemics [8]. This can be seen in the patterns found, thus highlighting the need for reliable online sources. HCPs likely seek information on oral suspension of oseltamivir to diminish errors in prescribing medication in children [22, 23]. While general search engines cannot characterize

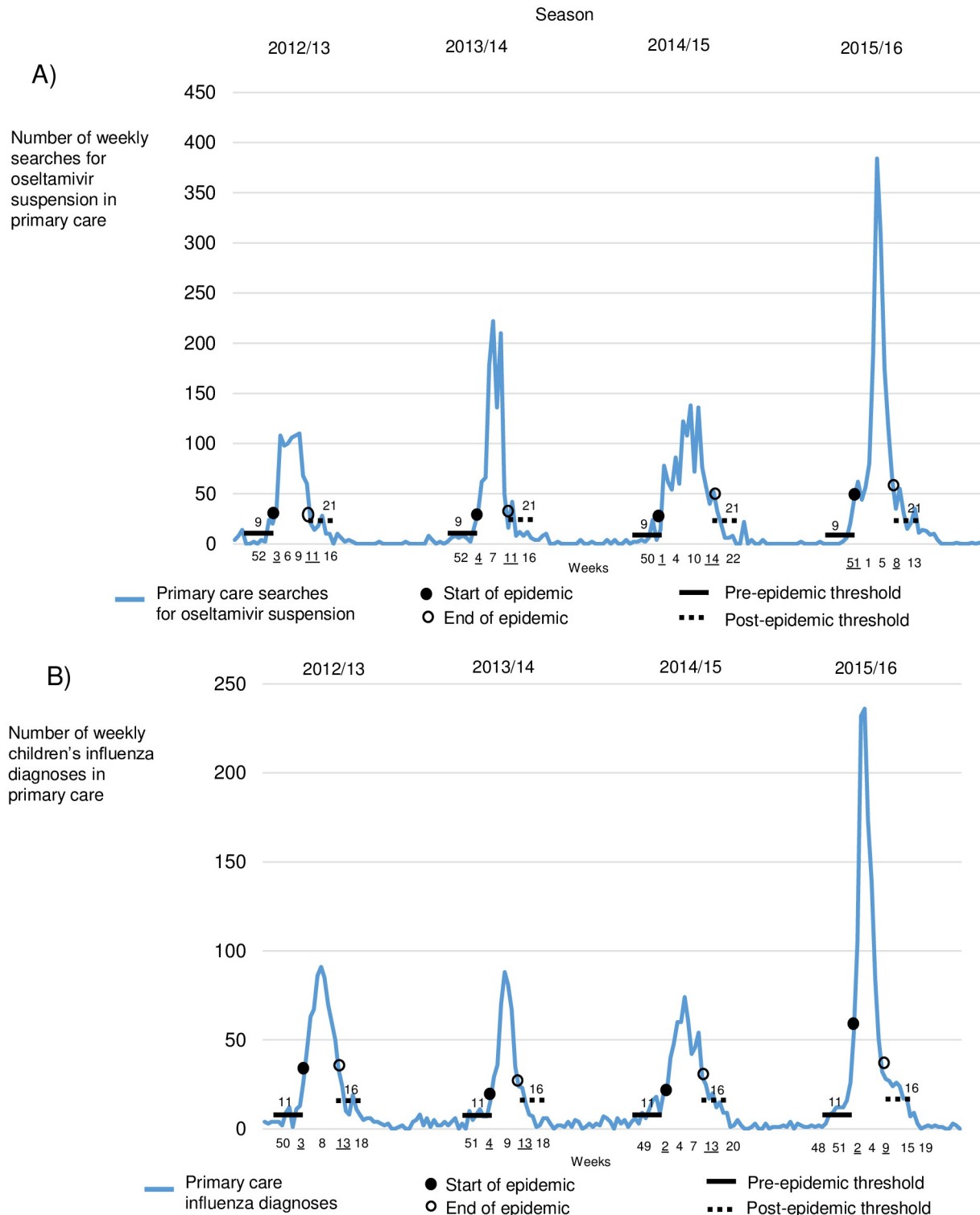

**Fig 2. Primary care searches, diagnoses, and starts and ends of influenza epidemic periods (underlined weeks) across Finland during 2012–2016 by season, including pre-epidemic and post-epidemic influenza thresholds.** (A) Weekly searches for oseltamivir suspension in primary care. (B) Weekly children's influenza diagnoses in primary care.

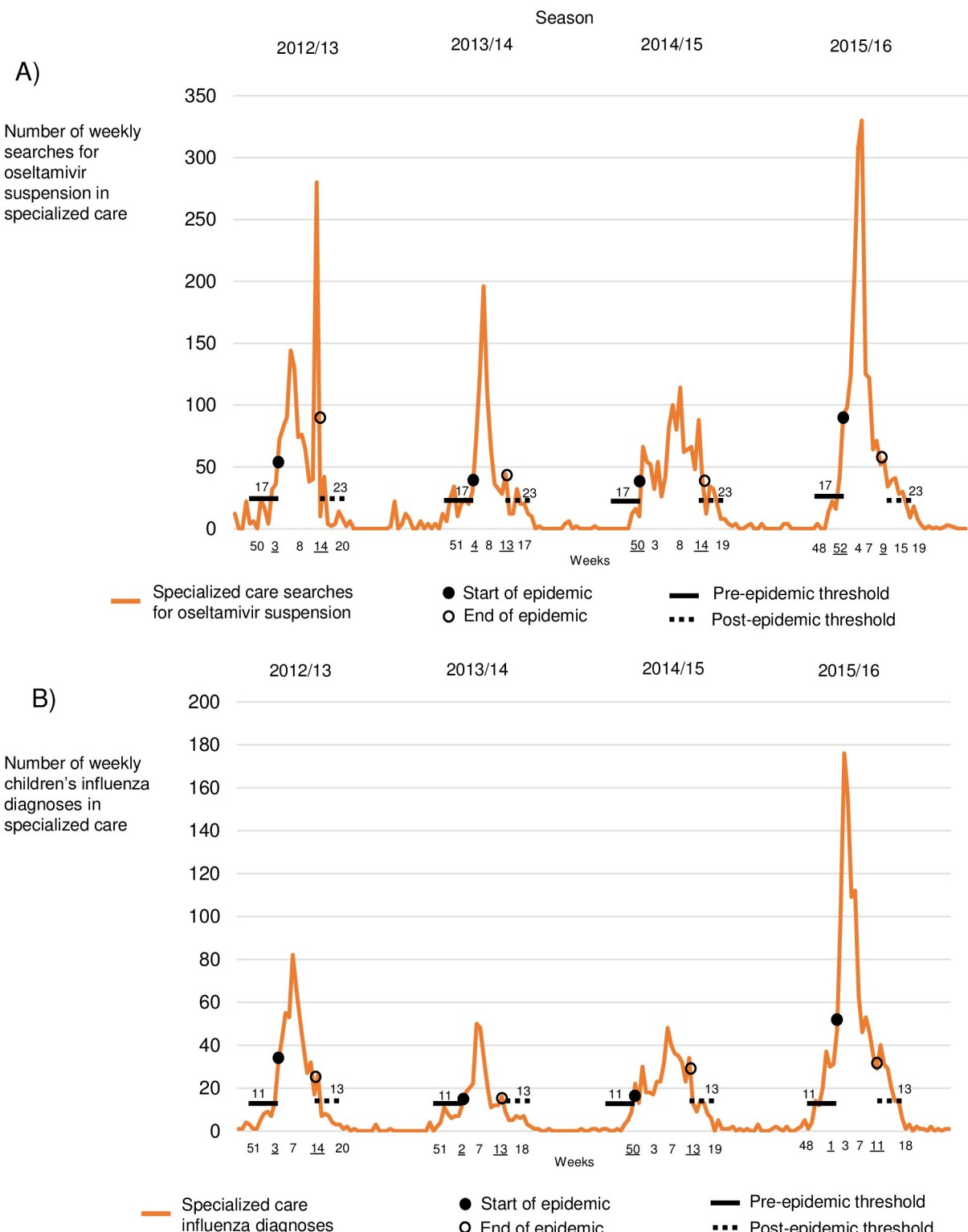

**Fig 3. Specialized care searches, diagnoses, and starts and ends of influenza epidemic periods (underlined weeks) across Finland during 2012–2016 by season, including pre-epidemic and post-epidemic influenza thresholds.** (A) Weekly searches for oseltamivir suspension in specialized care. (B) Weekly children's influenza diagnoses in specialized care.

**Table 2. MEM-calculated starts and ends of the epidemic seasons by searches for oseltamivir oral suspension and children's influenza diagnoses in primary and specialized care, and laboratory reports of children's influenza A and B in Finland 2012–2016.**

| | Primary care | | | | Specialized care | | | | Laboratory reports | | | |
|---|---|---|---|---|---|---|---|---|---|---|---|---|
| | Searches for oral suspension of oseltamivir | | Children's influenza diagnoses | | Searches for oral suspension of oseltamivir | | Children's influenza diagnoses | | Influenza A | | Influenza B | |
| | Epidemic starts | | Epidemic starts | | Epidemic starts | | Epidemic starts | | Epidemic starts | | Epidemic starts | |
| Season | Week | Date | Week | Date | Week | Date | Week | Date | Week | Date | Week | Date |
| 2012/13 | 3 | Jan 14 | 3 | Jan 14 | 3 | Jan 14 | 3 | Jan 14 | 3 | Jan 14 | 4 | Jan 21 |
| 2013/14 | 4 | Jan 20 | 4 | Jan 20 | 4 | Jan 20 | 2 | Jan 6 | 3 | Jan 13 | 6 | Feb 3 |
| 2014/15 | 1 | Dec 29 | 2 | Jan 5 | 50 | Dec 8 | 50 | Dec 8 | 50 | Dec 8 | 4 | Jan 19 |
| 2015/16 | 51 | Dec 14 | 2 | Jan 11 | 52 | Dec 21 | 1 | Jan 4 | 52 | Dec 21 | 6 | Feb 8 |
| | Primary care | | | | Specialized care | | | | Laboratory reports | | | |
| | Searches for oral suspension of oseltamivir | | Children's influenza diagnoses | | Searches for oral suspension of oseltamivir | | Children's influenza diagnoses | | Influenza A | | Influenza B | |
| | Epidemic ends | | Epidemic ends | | Epidemic ends | | Epidemic ends | | Epidemic ends | | Epidemic ends | |
| Season | Week | Date | Week | Date | Week | Date | Week | Date | Week | Date | Week | Date |
| 2012/13 | 11 | Mar 11 | 13 | Mar 25 | 14 | Apr 1 | 14 | Apr 1 | 14 | Apr 1 | 17 | Apr 22 |
| 2013/14 | 11 | Mar 10 | 13 | Mar 24 | 13 | Mar 24 | 13 | Mar 24 | 13 | Mar 24 | 17 | Apr 21 |
| 2014/15 | 14 | Mar 30 | 13 | Mar 23 | 14 | Mar 30 | 13 | Mar 23 | 13 | Mar 23 | 17 | Apr 20 |
| 2015/16 | 8 | Feb 22 | 9 | Feb 29 | 9 | Feb 29 | 11 | Mar 14 | 9 | Feb 29 | 18 | May 2 |

their users, searches have been used in influenza surveillance and detection (GFT) [8–11]. Combining flu-related information from online and traditional sources has showed accurate and real-time predictions in influenza surveillance [12–14]. Our study found that the dedicated online medical source, PD [25], aimed at HCPs working in public primary and specialized care, could provide real-time information to be used in daily practice and in surveillance to detect influenza.

The strengths of our study are that HCPs (representativeness) use dedicated professional databases on the Internet (log data, real-time) and that register data were available for comparing indicators of children's influenza. However, this study has certain limitations. A smaller amount of search- and register data of children's influenza compared to our previous study on HCP searches [24] may have an effect on the analyses and results. First, our study methodology may include some possible confounders that affect interpretation of the results of online searches, meaning that various HCPs in primary and specialized care may have searched for oseltamivir differently. In our study, we assumed that oral suspension of oseltamivir is mainly used in children. Some of these searches may have been done in connection with treatment of adult or elderly patients unable to swallow tablets, thus liquid oseltamivir is searched for instead. In specialized care, HCPs other than pediatricians may also have searched for information on oseltamivir, and some professionals may have used paper sources and consulted colleagues when verifying the correct dosage of oral suspension of oseltamivir, thus decreasing searches. It is also worth noting that some HCPs may be familiar with the correct oseltamivir dosage once verified, thus a database search is not performed, especially when a physician regularly encounters children with similar weights or ages. Some searches may have been performed by medical students for learning purposes or senior physicians in teaching situations. In addition, HCPs other than physicians (e.g., nurses, pharmacists) may have searched for information on oseltamivir. However, we assume that the majority of oseltamivir searches occur in practice in primary and specialized care by clinical physicians.

**Table 3. Pairs, paired differences with the mean, bias-corrected and accelerated confidence intervals and p-values, and correlations.** Primary care searches for oseltamivir oral suspension, specialized care searches for oseltamivir oral suspension, children's influenza diagnoses in primary and specialized care, and laboratory reports of children's influenza A and B were paired and calculated and bootstrapped according to epidemic starting week.

| Pair | Mean | Bias-corrected and accelerated 95% confidence interval of the mean (adjusted for ties) | | p-value of the mean | Kendall's correlation coefficient (τ) |
|------|------|------|------|------|------|
| | | Lower | Upper | | |
| Oseltamivir searches in primary care–Influenza diagnoses in primary care | −1.0 | −3.0 | −0.3 | 0.132 | 0.913 |
| Oseltamivir searches in specialized care–Influenza diagnoses in primary care | −1.5 | −4.0 | −0.5 | 0.124 | 0.913 |
| Oseltamivir searches in specialized care–Influenza A | 0.3 | 0.0 | 0.5 | 0.624 | 0.913 |
| Influenza A–Influenza diagnoses in specialized care | 0.0 | −1.0 | 0.5 | 0.001 | 0.913 |
| Influenza A–Influenza diagnoses in primary care | −1.8 | −3.5 | −0.8 | 0.006 | 0.800 |
| Oseltamivir searches in specialized care–Influenza diagnoses in specialized care | 0.3 | −0.8 | 1.3 | 0.880 | 0.667 |
| Oseltamivir searches in primary care–Oseltamivir searches in specialized care | 0.5 | −0.8 | 2.1 | 0.662 | 0.667 |
| Influenza diagnoses in specialized care–Influenza diagnoses in primary care | −1.8 | −4.0 | −0.8 | 0.010 | 0.548 |
| Oseltamivir searches in primary care–Influenza A | 0.8 | −0.5 | 2.0 | 0.404 | 0.548 |
| Oseltamivir searches in specialized care–Influenza B | −3.8 | −6.0 | −2.3 | 0.001 | 0.408 |
| Oseltamivir searches in primary care–Influenza diagnoses in specialized care | 0.8 | −1.5 | 2.3 | 0.474 | 0.333 |
| Influenza B–Influenza diagnoses in primary care | 2.3 | 1.3 | 3.3 | < 0.001 | 0.224 |
| Influenza A–Influenza B | −4.0 | −6.0 | −2.3 | 0.001 | 0.224 |
| Influenza B–Influenza diagnoses in specialized care | 4.0 | 1.8 | 5.3 | < 0.001 | 0.000 |
| Oseltamivir searches in primary care–Influenza B | −3.3 | −7.0 | −1.8 | 0.001 | 0.000 |

Second, HCPs and health care units may report diagnoses and test children differently. We found that influenza diagnoses in primary care appeared slightly higher compared to specialized care (3,717 vs. 2,837) since the searches in primary and specialized care appeared mainly similar (5,281 vs. 5,658) (Table 1). This could mean that HCPs encounter children with influenza and search for oseltamivir in both sectors, but due to the large number of public primary care units (health care centers) in Finland, which more children attend, primary care HCPs may diagnose more children with influenza. However, some specialized care HCPs may have reported children's influenza-like symptoms in a broader category of infectious diseases, such as acute respiratory infections. It is worth noting that no conclusions of sector differences in laboratory findings can be drawn since we could not distinguish children's laboratory findings of influenza A (4,518) and B (1,663) between primary and specialized care. The total number of diagnoses (6,554) is slightly higher than total number of laboratory findings (6,181). This means that every child presenting influenza-like symptoms is not tested for the virus (but diagnosed with influenza), especially in primary care where laboratory facilities may be poor quality. Although specialized care may be quicker to test children for the virus and thus find positive results more often, we found a smaller amount of influenza diagnoses in specialized care. There may be a wide variation of reporting diagnoses in distinct primary and specialized care units nationwide. In addition, some HCPs in primary and specialized care units may also report an influenza diagnosis incorrectly in the electronic patient record since laboratory findings have been properly transferred to NIDR by microbiological laboratories. In addition, the oseltamivir search data originated from across the country, including no geographical variations, thus influenza epidemics may begin in different regions of Finland at different times.

Third, while more likely to occur in the general population, the media coverage on the start of influenza season or other influenza-related news during the season may affect information

seeking behavior among HCPs, thus showing the increase in oseltamivir searches. However, we did not measure media coverage and its potential impact on searches. We have previously shown [33, 34] searches performed by HCPs to be more specific to recognized outbreaks than searches performed by the general population and therefore consider this to have a minor impact on the results. It is important to note that some parents may request a doctor to prescribe oseltamivir regardless of a test result. Due to these occasional cases, some searches do not accurately indicate true influenza epidemics, but epidemics of fear [11, 35]. This phenomenon may also increase the database searches for oseltamivir, possibly resulting in the biased real-time detection of infectious diseases, such as seasonal influenza outbreaks. However, we do not consider the impact of this on our results to be significant. The databases (PD, the national register of primary and specialized care diagnoses, and NIDR) are separate, thus no searches, diagnoses, and laboratory findings can be connected to one another or the same patient.

## Conclusion

To our knowledge, this is the first study to compare HCPs' online searches for children's oseltamivir in public primary and specialized care to influenza diagnoses. We found that similar patterns could be seen in oral suspension of oseltamivir searches and children's influenza diagnoses in primary and specialized health care in Finland during 2012–2016. Searches and diagnoses mainly appeared in the same week during influenza season, but searches preceded diagnoses only sometimes. This emphasizes not only use of online database searches in infectious disease surveillance, but also highlighting various HCPs' similar searching behavior in health care sectors. The possibility of dosage errors in prescribing medication in children highlights the need for reliable online databases aimed at HCPs who seek medication information. HCPs' searches for children's oseltamivir could be used as an additional source of information for disease surveillance when detecting influenza epidemics. It is important to study and understand the characteristics affecting searching behavior to benefit various information sources in the future. Further studies should assess statistical analyses skewed toward the prediction power of the oseltamivir search in actual influenza cases. Future research is needed to focus on the applicability of the results in different infectious diseases, as well as in other health care sectors and medical databases in other countries.

## Supporting information

**S1 Data.**
(XLSX)

## Acknowledgments

Preliminary results from this study were presented at the ESPID (European Society for Paediatric Infectious Diseases) conference, May 28–June 2, 2018, Malmö, Sweden.

## Author Contributions

**Conceptualization:** Milla Mukka, Samuli Pesälä, Aapo Juutinen, Mikko J. Virtanen, Pekka Mustonen, Minna Kaila, Otto Helve.

**Data curation:** Milla Mukka, Samuli Pesälä, Aapo Juutinen, Mikko J. Virtanen, Otto Helve.

**Formal analysis:** Milla Mukka, Samuli Pesälä, Aapo Juutinen, Mikko J. Virtanen.

**Methodology:** Aapo Juutinen, Mikko J. Virtanen.

**Project administration:** Milla Mukka, Samuli Pesälä, Pekka Mustonen, Minna Kaila, Otto Helve.

**Software:** Aapo Juutinen.

**Supervision:** Samuli Pesälä, Minna Kaila, Otto Helve.

**Writing – original draft:** Milla Mukka, Samuli Pesälä.

**Writing – review & editing:** Milla Mukka, Samuli Pesälä, Aapo Juutinen, Mikko J. Virtanen, Pekka Mustonen, Minna Kaila, Otto Helve.

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
