## [Decision Letter · Decision Letter 0]

20 Oct 2021

PONE-D-21-11786

Healthcare professionals’ online searches of children’s oseltamivir in primary and specialized care

PLOS ONE

Dear Dr. Mukka,

Thank you for submitting your manuscript to PLOS ONE. After careful consideration, we feel that it has merit but does not fully meet PLOS ONE’s publication criteria as it currently stands. Therefore, we invite you to submit a revised version of the manuscript that addresses the points raised during the review process.

The manuscript has been evaluated by two reviewers, and their comments are available below.

The reviewers have raised a number of concerns that need attention. They request additional information on the utility of this data in assessment of influenza severity, the non-qualitative assessment methods used in this study, sources of information for searches, research objectives/rationale, and various other moderate comments on manuscript structure and discussion.

Could you please revise the manuscript to carefully address the concerns raised?

We look forward to receiving your revised manuscript.

Kind regards,

Sebastian Shepherd

Associate Editor

PLOS ONE

Journal Requirements:

2. Thank you for stating the following in the Competing Interests section: "MK has held various trustee positions in the Finnish Medical Society Duodecim since the late 1990s. OH has held various trustee positions in the Finnish Medical Society Duodecim and Duodecim Medical Publications Ltd since 2009 and is a partner at iHealth Finland Ltd. The other authors have no competing interests." 

Reviewers' comments:

Reviewer's Responses to Questions

**Comments to the Author**

1. Is the manuscript technically sound, and do the data support the conclusions?

Reviewer #1: Partly

Reviewer #2: Partly

2. Has the statistical analysis been performed appropriately and rigorously? 

Reviewer #1: No

Reviewer #2: No

3. Have the authors made all data underlying the findings in their manuscript fully available?

Reviewer #1: Yes

Reviewer #2: Yes

4. Is the manuscript presented in an intelligible fashion and written in standard English?

Reviewer #1: Yes

Reviewer #2: Yes

5. Review Comments to the Author

Reviewer #1: There is always a need for new sources of influenza and respiratory virus surveillance data. This study provides evidence of generally good alignment between physician searches related to oseltamivir use in children and other influenza surveillance sources based purely on visual review of time series data. My overarching comments are the following. The authors should make clearer exactly how and in what circumstances this data type may be informative for influenza severity– e.g., to assess severity. I also think some of the results that they highlight are not particularly interesting and it would be best to focus on recommendations for using these data. Their prior work (ref 18) used quantitative methods to assess similar data – it is not clear why they have not employed those methods in this study. Other specific comments are below.

Abstract

I recommend highlighting here that the intent of the study is to assess whether these search data can be used for surveillance purposes. I believe that is the objective of the study and that the text of the objective and rationale sections should be swapped.

The absolute numbers of searches vs diagnoses is not the most important results to highlight. I also don’t think primary vs specialized care based searches are key findings to include in the abstract – unless you make clear that specialized includes hospital care (see comment below).

Introduction

Some editing of the introduction would make the information flow better – either rearrange so influenza and oseltamivir is introduced first, or add paragraph breaks. Right now the first paragraph is muddled.

Lines 51-52 – Seems unnecessary to mention a company (Google) three times – would instead just refer to Google Flu Trends once.

Lines 55-56 – Could be omitted as this is well understood background information.

Since the introduction explains that “specialized care” includes hospital/inpatient based care, and “primary care” seems to only be ambulatory/outpatient, consider calling the specialized care category “specialized and hospital care”. That is an important distinction and makes interpreting the results from the two categories more meaningful.

Much of the information in lines 60-83 should be in the Methods section.

Methods

There are no quantitative methods applied in the study. Visual examination of trends and general comparisons of counts of data is all that is presented, yet they have conducted what appears to be a very similar study previously (ref #18) and applied quite different methods.

Results

Figures 1 and 2 can be combined.

Line 114 – 115 – I’m not sure why a “higher” peak for searches, based on counts, compared to other data sources is important to highlight. Timing of the different data sources is key, as well as what the relative trends are for primary vs specialized and hospital based care – i.e., severity of the season. If the specialized searches for oseltamivir start to appear sooner than other indicators of activity, that is important to highlight. Commenting on what is known about the severity of the seasons you studied would also be useful – to help readers understand how such data can help elucidate influenza activity per season, in “real time”.

The peak of specialized searches in the spring of the 2013-14 season does not align with diagnoses or lab reports – what is your hypothesis? This is mentioned in the results but never discussed. This may just be a data anomaly but it is quite evident and needs to be addressed in some way.

Reviewer #2: Dear editor,

Thank you for providing me with this opportunity to review the mmanuscript titled “Healthcare professionals’ online searches of children’s oseltamivir in primary and specialized care” submitted to Plos One journal.

In their interesting work, authors have focused on trying to establish a relationship between the number of searches in the primary and specialized care for oseltamivir and the recorded cases of flu cases in children. This study is valuable because the findings can potentially contribute to the effective prediction of flu pandemics using the frequency of searches for a specific term/phrase.

I believe that this manuscript can be considered for further review with additional work, and I do not suggest publication in its current form. I have made detailed comments related to each section of the manuscript below. This manuscript presents a lack of rigorous statistical analysis and an underdeveloped methodology, which need special attention.

Title:

Suggestions: Change the title to something more generalizable that shows the usefulness of online search results for flu outbreak detection/prediction

Keywords

Correct “speralized care” to “Specialized care”

Abstract

Please consider these comments for the abstract and apply those to the body of the manuscript wherever applicable.

Well-constructed and organized, overall. Important notes:

Lines 24-30: The introductory sentences in the “Objective” section do not align correctly with the research objective. More specifically, the last sentence needs to be changed to reflect the objective of this research (closer to the “Rationale”).

Lines 34-38: Include the source of information/ data for both online searches and the recorded number of cases.

Lines 39-44: Results need some rigorous statistical analysis. At least, the correlation between the number of searches and the number of recorded cases should be considered. Moreover, if possible additional variables for regression analysis are suggested. Probably a formula can be developed to predict the number of cases using the frequency of searches.

Lines 45-46: The conclusion is stated in a form that suggests a binary outcome/incident as yes/no. It should be rephrased as a directional association. More searches= more reported cases (A positive correlation between the frequency of searches and the # of reported cases)

Introduction

Overall the content of the introduction section is satisfactory. However, the organization and flow are relatively heavy to digest. Please consider using different paragraphs for each of the following content in the following order:

1- Influenza (include # of annual cases in Finland (or Prevalence rate), deaths, plus the burden of disease in terms of cost to the healthcare system.

2- Different options for Online searches, pros cons.

3- Details about the setting (Finish healthcare system) relevant to the topic.

4- The rationale for the need for this research and its expected value added to literature/practice.

Material and Methods

This section is poorly developed and need much work, including:

1- Suggest adding/ or moving some content from the introduction and adding a “Setting” sub-heading to discuss the study setting.

2- The data sources for online searches and recorded cases should be included here.

3- The rationale and logic behind choosing this medication (oseltamivir) and why this one(and not other medications/ keywords?) should be clarified.

4- Important: The analysis is insufficient. At least, the correlation between the number of searches and the number of recorded cases should be considered. Moreover, if possible additional variables for regression analysis are suggested. Probably a formula can be developed to predict the number of cases using the frequency of searches.

Line 102: change “laboratory positive findings” to “positive laboratory findings”

Results

The results section needs additional work.

Important: As a result of substantial changes suggested to the “Methods” section, the “Results” section also needs more development. Begin by including the descriptive statistics about the # of searches. Then, expand with correlation/regression findings. Some charts can be useful here

Line 114 and elsewhere: The season 2012/13 needs to be written transparently. For example, the phrase “In specialized care 2012/13 and 2013/14,” is confusing. Please use year and season to make these statements easy to understand.

Line 115: Please rewrite this phrase “a wider scale of months before and after these seasons”. It’s a bit unclear.

Discussion

The discussion section talks about the potential reasons behind why searches are more/less in primary/specialized settings and probably why the reported cases are different in these settings. However, there is no discussion about the uses of these search result for prediction and potential confounders that can impact the accuracy/ issues of using these search results to predict the number of cases. Moreover, other studies/ applications similar to this research should be included and discussed here.

Lines 140-141: This statement does not reflect the relative size of the primary and specialized care. The total number of practitioners in different levels (primary/specialized care) should be included to make a legitimate comparison. More can be discussed here by knowing the size of each care level. Needs to be corrected “However, our study found that professionals in primary care use PD as much as their colleagues in specialized care”

Conclusions

Change based on the results from additional statistical analysis leaned toward the prediction power of the search for oseltamivir in actual flu cases.

The suggestion for future research in other diseases is proper.

6. PLOS authors have the option to publish the peer review history of their article (what does this mean?). If published, this will include your full peer review and any attached files.

Reviewer #1: No

Reviewer #2: **Yes: **Ahmad Khanijahani

---

## [Author Response · Author response to Decision Letter 0]

24 Jan 2022

Reviewer #1: There is always a need for new sources of influenza and respiratory virus surveillance data. This study provides evidence of generally good alignment between physician searches related to oseltamivir use in children and other influenza surveillance sources based purely on visual review of time series data. My overarching comments are the following. The authors should make clearer exactly how and in what circumstances this data type may be informative for influenza severity– e.g., to assess severity. I also think some of the results that they highlight are not particularly interesting and it would be best to focus on recommendations for using these data. Their prior work (ref 18) used quantitative methods to assess similar data – it is not clear why they have not employed those methods in this study. Other specific comments are below.

Response: Thank you for your general comment on our study.

Abstract

I recommend highlighting here that the intent of the study is to assess whether these search data can be used for surveillance purposes. I believe that is the objective of the study and that the text of the objective and rationale sections should be swapped.

Response: Thank you for your comment. We have now used the following sections in Abstract: Introduction, Methods, Results, and Conclusion. Also, we have revised the objective and rational as suggested, and included them in the Introduction section. We also embedded further information to introduce the study background. In Abstract/Conclusion, we mention that “…the searches should be considered as a supplementary source of information for disease surveillance when detecting influenza epidemics.” The Abstract section is also revised due to the words allowed (max 300 words).

The absolute numbers of searches vs diagnoses is not the most important results to highlight. I also don’t think primary vs specialized care based searches are key findings to include in the abstract – unless you make clear that specialized includes hospital care (see comment below).

Response: Thank you for this comment. We have now used statistical analyses in our study, and added statistical data in Abstract/Results. Also, hospitals are now mentioned in Abstract.

Introduction

Some editing of the introduction would make the information flow better – either rearrange so influenza and oseltamivir is introduced first, or add paragraph breaks. Right now the first paragraph is muddled.

Response: Thank you for your suggestion. We have now rearranged Introduction and divided it into four paragraphs: influenza/oseltamivir, online searching, Finnish healthcare system/professionals, and rationale/objective. 

Lines 51-52 – Seems unnecessary to mention a company (Google) three times – would instead just refer to Google Flu Trends once.

Response: We have omitted Google (mentioned twice) as suggested. We mention GFT only.

Lines 55-56 – Could be omitted as this is well understood background information.

Response: Thank you for this suggestion. However, we have now reorganized the paragraphs of the Introduction section with additional information on influenza, and therefore we have left this sentence in the text.

Since the introduction explains that “specialized care” includes hospital/inpatient based care, and “primary care” seems to only be ambulatory/outpatient, consider calling the specialized care category “specialized and hospital care”. That is an important distinction and makes interpreting the results from the two categories more meaningful.

Response: Thank you for your important comment. In Finland, public specialized care is comprised of both outpatient clinics and hospitals, and our data on specialized care include both. The point is that specialists (such as pediatricians) work in public specialized care, while general practitioners work in public primary care (such as in healthcare centers). In Finland, the term “hospital” comprises outpatient/inpatient in distinct specialty, meaning that a specialist works in outpatient clinics encountering patients within their own specialty, for example in pediatrics. We mention in Introduction: “Pediatricians in specialized care work in emergencies, outpatient clinics, and hospital wards…”

Much of the information in lines 60-83 should be in the Methods section.

Response: We have now relocated much of the information from Introduction to Methods as suggested. Also, Introduction is now reorganized by dividing it into four paragraphs.

Methods

There are no quantitative methods applied in the study. Visual examination of trends and general comparisons of counts of data is all that is presented, yet they have conducted what appears to be a very similar study previously (ref #18) and applied quite different methods.

Response: Thank you for your comment. We have now added statistical analyses in our study and described it in Materials and Methods. We used the Moving Epidemic Method (MEM) to calculate the starting and ending weeks with pre-epidemic and post-epidemic thresholds. The MEM model is the same method we used in our previous study (ref #18). Also, paired differences with correlations were assessed.

Results

Figures 1 and 2 can be combined.

Response: Thank you for your suggestion. Figures were meant to be combined during the layout processing, so that they are set in one figure in order to be seen at a glance. Also, more figures have been added in order to be set into three panels (Figures 1A-B, Figures 2A-B, and Figures 3A-B).

Line 114 – 115 – I’m not sure why a “higher” peak for searches, based on counts, compared to other data sources is important to highlight. Timing of the different data sources is key, as well as what the relative trends are for primary vs specialized and hospital based care – i.e., severity of the season. If the specialized searches for oseltamivir start to appear sooner than other indicators of activity, that is important to highlight. Commenting on what is known about the severity of the seasons you studied would also be useful – to help readers understand how such data can help elucidate influenza activity per season, in “real time”.

Response: Thank you for pointing out this important detail. It is true that timing is key in terms of comparing different data. However, higher peaks for searches (done by healthcare professionals) may mirror different information seeking behaviors among primary and specialized care professionals, not just comparing data between indicators, but also between professionals working in different healthcare sectors (specialized care pediatricians vs. primary care general practitioners; both encountering children with influenza and prescribing oral suspension of oseltamivir [In Finland, this is also legitimated for general practitioners]). In this study, we wanted to showcase how searches, influenza diagnoses, and laboratory findings could coincide with one another. Along with this, the number of professionals’ searches in primary and specialized care was the aim of our research, and thus we found it important to study. However, we have now omitted information on peaks.

Of note, we have run statistical analyses from our data in order to highlight the timing of different indicators. We used the Moving Epidemic Method (MEM model) to calculate the starting and ending weeks of an epidemic with pre-epidemic and post-epidemic thresholds. Paired differences were assessed as well.

The peak of specialized searches in the spring of the 2013-14 season does not align with diagnoses or lab reports – what is your hypothesis? This is mentioned in the results but never discussed. This may just be a data anomaly but it is quite evident and needs to be addressed in some way.

Response: Thank you for your comment. In Figure 2, three peaks are shown during the 2013/14 season (February-March). Searches are align with laboratory reports (influenza A) (searches peak higher instead), but diagnoses stay behind. However, we have omitted information on peaks, because additional statistical analyses were added into this study.

Reviewer #2: Dear editor,

Thank you for providing me with this opportunity to review the manuscript titled “Healthcare professionals’ online searches of children’s oseltamivir in primary and specialized care” submitted to Plos One journal.

In their interesting work, authors have focused on trying to establish a relationship between the number of searches in the primary and specialized care for oseltamivir and the recorded cases of flu cases in children. This study is valuable because the findings can potentially contribute to the effective prediction of flu pandemics using the frequency of searches for a specific term/phrase.

I believe that this manuscript can be considered for further review with additional work, and I do not suggest publication in its current form. I have made detailed comments related to each section of the manuscript below. This manuscript presents a lack of rigorous statistical analysis and an underdeveloped methodology, which need special attention.

Response: Thank you for your general comment on our study.

Title:

Suggestions: Change the title to something more generalizable that shows the usefulness of online search results for flu outbreak detection/prediction

Response: Thank you for this suggestion. We have now revised the title as follows: “Online searches of children’s oseltamivir in public primary and specialized care - Detecting influenza outbreaks in Finland using dedicated databases for health care professionals.”

Keywords:

Correct “speralized care” to “Specialized care”

Response: Sorry, a typo. We have corrected this keyword. 

Abstract:

Please consider these comments for the abstract and apply those to the body of the manuscript wherever applicable. 

Well-constructed and organized, overall. Important notes:

Lines 24-30: The introductory sentences in the “Objective” section do not align correctly with the research objective. More specifically, the last sentence needs to be changed to reflect the objective of this research (closer to the “Rationale”).

Response: Thank you for your comment. We have now used the following sections in Abstract: Introduction, Methods, Results, and Conclusion. Also, we have revised the objective and rational as suggested, and included them in the Introduction section. We have also embedded further information to introduce study background.

Lines 34-38: Include the source of information/ data for both online searches and the recorded number of cases.

Response: Thank you for this comment. We have added the information sources as suggested. 

Lines 39-44: Results need some rigorous statistical analysis. At least, the correlation between the number of searches and the number of recorded cases should be considered. Moreover, if possible additional variables for regression analysis are suggested. Probably a formula can be developed to predict the number of cases using the frequency of searches. 

Response: We have now added a rigorous statistical analysis (Moving Epidemic Method (MEM)) in our study. Also, paired differences with correlations were calculated.

Lines 45-46: The conclusion is stated in a form that suggests a binary outcome/incident as yes/no. It should be rephrased as a directional association. More searches= more reported cases (A positive correlation between the frequency of searches and the # of reported cases)

Response: Thank you for your comment. We have now rephrased the conclusion sections of Abstract and Conclusion according to our results/findings we conclude. Correlations calculated from MEM results (starting weeks) are mentioned in Abstract and Results.

Introduction:

Overall the content of the introduction section is satisfactory. However, the organization and flow are relatively heavy to digest. Please consider using different paragraphs for each of the following content in the following order:

Response: Thank you for your comment. We have now revised the Introduction section by dividing the section into four paragraphs as suggested. 

1- Influenza (include # of annual cases in Finland (or Prevalence rate), deaths, plus the burden of disease in terms of cost to the healthcare system.

Response 1: Thank you for this important comment. We have now added a couple of sentences about these issues you suggest. Unfortunately, we have no numerical data available on deaths and costs of children’s influenza in Finland. However, we have described the burden of influenza on the society and how it affects healthcare system and families. Also, we found two studies on incidence of influenza in Finnish children, and we described the results in the first paragraph of Introduction. 

2- Different options for Online searches, pros cons.

Response 2: We have now added different options for online searches and described the challenges with additional literature embedded. 

3- Details about the setting (Finish healthcare system) relevant to the topic.

Response 3: The Finnish healthcare system (details about public primary and specialized care) is introduced here. 

4- The rationale for the need for this research and its expected value added to literature/practice.

Response 4: We have added the aim and rationale of this study, and the expected value to added literature/practice.

Material and Methods:

This section is poorly developed and need much work, including:

1. Suggest adding/ or moving some content from the introduction and adding a “Setting” sub-heading to discuss the study setting. 

Response 1: We have relocated some content from the Introduction section to Material and Methods and added the “Setting” subheading as suggested. Please see below (number 2).

2. The data sources for online searches and recorded cases should be included here.

Response 2: We mention the data sources for online searches (Physician’s Databases, PD) and for recorded cases (National Infectious Disease Register, NIDR) in Materials and Methods/Setting. We have now relocated the content (details on PD and NIDR), from the Introduction section as suggested. 

3. The rationale and logic behind choosing this medication (oseltamivir) and why this one (and not other medications/ keywords?) should be clarified.

Response 3: Thank you for your comment. We have now clarified in the “Setting” section why we chose oseltamivir in our study. 

4. Important: The analysis is insufficient. At least, the correlation between the number of searches and the number of recorded cases should be considered. Moreover, if possible additional variables for regression analysis are suggested. Probably a formula can be developed to predict the number of cases using the frequency of searches.

Response 4: We have added statistical analyses in our study. We used the Moving Epidemic Method (MEM model) to calculate the starting and ending weeks of an epidemic with pre-epidemic and post-epidemic thresholds. Also, correlations were calculated (paired differences). We have also added the “Descriptive and Statistical Analysis” subheading in Materials and Methods. 

Line 102: change “laboratory positive findings” to “positive laboratory findings”

Response: We have changed this as suggested. 

Results:

The results section needs additional work. 

Important: As a result of substantial changes suggested to the “Methods” section, the “Results” section also needs more development. Begin by including the descriptive statistics about the # of searches. Then, expand with correlation/regression findings. Some charts can be useful here

Response: Thank you for your suggestion. We have now reorganized the Results section (by beginning the number of searches) as suggested. Statistical analyses, including the Moving Epidemic Method (MEM), have been added and the results tabled.

Line 114 and elsewhere: The season 2012/13 needs to be written transparently. For example, the phrase “In specialized care 2012/13 and 2013/14,” is confusing. Please use year and season to make these statements easy to understand.

Response: We have revised years and seasons as suggested. However, we have omitted information on seasons due to changes in analyses used in this study (MEM). 

Line 115: Please rewrite this phrase “a wider scale of months before and after these seasons”. It’s a bit unclear.

Response: Due to statistical analyses added in our study (MEM), we have omitted the sentence as being unnecessary.

Discussion:

The discussion section talks about the potential reasons behind why searches are more/less in primary/specialized settings and probably why the reported cases are different in these settings. However, there is no discussion about the uses of these search result for prediction and potential confounders that can impact the accuracy/ issues of using these search results to predict the number of cases. Moreover, other studies/ applications similar to this research should be included and discussed here.

Response: Thank you for this comment. We have described the limitations of this study and also added more information on confounders that may impact the results of our study (searches), also from the view of predicting/detecting the number of cases. We have also included other studies similar to this research, and some sentences are relocated in the paragraph of limitations.

Lines 140-141: This statement does not reflect the relative size of the primary and specialized care. The total number of practitioners in different levels (primary/specialized care) should be included to make a legitimate comparison. More can be discussed here by knowing the size of each care level. Needs to be corrected “However, our study found that professionals in primary care use PD as much as their colleagues in specialized care”

Response: Thank you for your important comment. This is true that the statement does not reflect the relative size of primary and specialized care. We have now omitted the statement since we have no accurate information on the professionals working in primary and specialized care during those years.

Conclusions:

Change based on the results from additional statistical analysis leaned toward the prediction power of the search for oseltamivir in actual flu cases. The suggestion for future research in other diseases is proper.

Response: We have added that future studies should use additional statistical analyses leaned toward the prediction power of the search of oseltamivir in actual influenza cases.

---

## [Decision Letter · Decision Letter 1]

28 Apr 2022

PONE-D-21-11786R1Online searches of children’s oseltamivir in public primary and specialized care ― Detecting influenza outbreaks in Finland using dedicated databases for health care professionalsPLOS ONE

Dear Dr. Mukka,

Thank you for submitting your manuscript to PLOS ONE. After careful consideration, we feel that it has merit but does not fully meet PLOS ONE’s publication criteria as it currently stands. Therefore, we invite you to submit a revised version of the manuscript that addresses the points raised during the review process.

We have the response from the reviewers assigned for your revised manuscript. Overall, reviewers indicated that the revisions made by the authors addressed most of the questions raised by the first two reviewers. Moreover, I reviewed your revised manuscript, and I believe that with minor revisions, it can be considered for publication

Please consider the points made by the new reviewer (#4) and submit a revised version of your manuscript. Please also double-check your manuscript for the English language and proofread it thoroughly.

We look forward to receiving your revised manuscript.

Kind regards,

Ahmad Khanijahani

Guest Editor

PLOS ONE

Journal Requirements:

Reviewers' comments:

Reviewer's Responses to Questions

**Comments to the Author**

1. If the authors have adequately addressed your comments raised in a previous round of review and you feel that this manuscript is now acceptable for publication, you may indicate that here to bypass the “Comments to the Author” section, enter your conflict of interest statement in the “Confidential to Editor” section, and submit your "Accept" recommendation.

Reviewer #3: (No Response)

Reviewer #4: All comments have been addressed

2. Is the manuscript technically sound, and do the data support the conclusions?

Reviewer #3: (No Response)

Reviewer #4: Partly

3. Has the statistical analysis been performed appropriately and rigorously? 

Reviewer #3: (No Response)

Reviewer #4: Yes

4. Have the authors made all data underlying the findings in their manuscript fully available?

Reviewer #3: (No Response)

Reviewer #4: Yes

5. Is the manuscript presented in an intelligible fashion and written in standard English?

Reviewer #3: (No Response)

Reviewer #4: No

6. Review Comments to the Author

Reviewer #3: (No Response)

Reviewer #4: This article provides the use of prescription antiviral dose look-up as a novel method for surveillance of influenza in children, and is an important and potentially helpful use of understanding influenza in our young populations.

**There could be more caveats provided around the possible confounders in these methodology: what other conclusions could be drawn from these data? are there any biases which arise from these data which have not been shared? and if so , what direction do they draw the results in? For example the difference in 1.7 in lab diagnostics between settings, but the dose checking was the same- what does this mean? what other explanations are there for this? Could anyone be looking up the does of osteltamivir for other reasons (ie teaching/learning?) Could people be prescribing without looking up the does once they know it?**

**The abstract doesn't read properly, and could do with checking for basic English.**

**However, this is a novel and important finding, and useful for surveillance of influenza in children, and is useful to share as helpful methods for public health practice.**

7. PLOS authors have the option to publish the peer review history of their article (what does this mean?). If published, this will include your full peer review and any attached files.

Reviewer #3: No

Reviewer #4: No

---

## [Author Response · Author response to Decision Letter 1]

24 May 2022

REBUTTAL LETTER

Academic editor: We have the response from the reviewers assigned for your revised manuscript. Overall, reviewers indicated that the revisions made by the authors addressed most of the questions raised by the first two reviewers. Moreover, I reviewed your revised manuscript, and I believe that with minor revisions, it can be considered for publication

Please consider the points made by the new reviewer (#4) and submit a revised version of your manuscript. Please also double-check your manuscript for the English language and proofread it thoroughly.

Response: Thank you for your comment. We have answered the points made by the reviewer #4 (please see below). The text has been language-checked, edited, and proofread in terms of proper English language. 

Academic editor: Please review your reference list to ensure that it is complete and correct. If you have cited papers that have been retracted, please include the rationale for doing so in the manuscript text, or remove these references and replace them with relevant current references. Any changes to the reference list should be mentioned in the rebuttal letter that accompanies your revised manuscript. If you need to cite a retracted article, indicate the article’s retracted status in the References list and also include a citation and full reference for the retraction notice.

Response: We have checked the Reference list to ensure that it is complete and correct. Due to additional information embedded in Discussion, three references [33,34,35] have been added on the Reference list. 

Reviewers' comments

Reviewer's Responses to Questions

Comments to the Author

1. If the authors have adequately addressed your comments raised in a previous round of review and you feel that this manuscript is now acceptable for publication, you may indicate that here to bypass the “Comments to the Author” section, enter your conflict of interest statement in the “Confidential to Editor” section, and submit your "Accept" recommendation.

Reviewer #3: (No Response)

Reviewer #4: All comments have been addressed

Response 1: Thank you for your comment.

2. Is the manuscript technically sound, and do the data support the conclusions?

Reviewer #3: (No Response)

Reviewer #4: Partly

Response 2: We have now added more information on conclusions supported by the data in the Discussion section. Please see below (Response 6).

3. Has the statistical analysis been performed appropriately and rigorously?

Reviewer #3: (No Response)

Reviewer #4: Yes

Response 3: Thank you for your comment. 

4. Have the authors made all data underlying the findings in their manuscript fully available?

Reviewer #3: (No Response)

Reviewer #4: Yes

Response 4: Thank you for your comment. 

5. Is the manuscript presented in an intelligible fashion and written in standard English?

Reviewer #3: (No Response)

Reviewer #4: No

Response 5: Thank you for pointing out your concern. The text has been language-checked, edited, and proofread in terms of proper English language.

6. Review Comments to the Author

Reviewer #3: (No Response)

Reviewer #4: This article provides the use of prescription antiviral dose look-up as a novel method for surveillance of influenza in children, and is an important and potentially helpful use of understanding influenza in our young populations.

There could be more caveats provided around the possible confounders in these methodology: what other conclusions could be drawn from these data? are there any biases which arise from these data which have not been shared? and if so , what direction do they draw the results in? For example the difference in 1.7 in lab diagnostics between settings, but the dose checking was the same- what does this mean? what other explanations are there for this? Could anyone be looking up the does of osteltamivir for other reasons (ie teaching/learning?) Could people be prescribing without looking up the does once they know it?

The abstract doesn't read properly, and could do with checking for basic English.

However, this is a novel and important finding, and useful for surveillance of influenza in children, and is useful to share as helpful methods for public health practice.

General response 6: Thank you for your comments. We have now added more information regarding your suggestions and answered them (and added in the manuscript) in detail below. Also, we have restructured the Discussion section due to the additional information embedded.

Comment: What other conclusions could be drawn from these data? Are there any biases which arise from these data which have not been shared? And if so, what direction do they draw the results in? 

Response: Thank you for your important comment. We have gone through the discussion among co-authors, and we believe that the confounders and biases are now described in the Discussion section. We have now added the following sentences about these possible confounders in our study:

“In addition, HCPs other than physicians (e.g., nurses, pharmacists) may have searched for information on oseltamivir. However, we assume that the majority of oseltamivir searches occur in practice in primary and specialized care by clinical physicians.”

“There may be a wide variation of reporting diagnoses in distinct primary and specialized care units nationwide.”

“Third, while more likely to occur in the general population, the media coverage on the start of influenza season or other influenza-related news during the season may affect information seeking behavior among HCPs, thus showing the increase in oseltamivir searches. However, we did not measure media coverage and its potential impact on searches.”

“It is important to note that some parents may request a doctor to prescribe oseltamivir regardless of a test result. Due to these occasional cases, some searches do not accurately indicate true influenza epidemics, but epidemics of fear [11,35]. This phenomenon may also increase the database searches for oseltamivir, possibly resulting in the biased real-time detection of infectious diseases, such as seasonal influenza outbreaks. However, we do not consider the impact of this on our results to be significant.”

Comment: For example the difference in 1.7 in lab diagnostics between settings, but the dose checking was the same- what does this mean? What other explanations are there for this?

Response: Thank you for your comment. We have re-checked the absolute numbers of diagnoses in primary and specialized care, and there are typos in the specialized care diagnoses (Table 1). Diagnoses in years 2012-16 equal 2,837 (not 6,181). Specialized care diagnoses in the whole column are the sum of influenza A and B columns (this is incorrect). These have now been revised. However, the MEM analysis has been run by using the corrected data as can be seen in Figures 1-3 (number of diagnoses on y axes similar). This typo in Table 1 does not affect the MEM results and MEM conclusions, but we have now revised the Discussion section that specialized care diagnoses would have appeared too high (6,181) compared to primary care (3,717). New conclusions have been added. We thank the reviewer for rigorous evaluation of the text.

We have revised the text as follows: “We found that influenza diagnoses in primary care appeared slightly higher compared to specialized care (3,717 vs. 2,837) since the searches in primary and specialized care appeared mainly similar (5,281 vs. 5,658) (Table 1). This could mean that HCPs encounter children with influenza and search for oseltamivir in both sectors, but due to the large number of public primary care units (health care centers) in Finland, which more children attend, primary care HCPs may diagnose more children with influenza.”

Comment: Could anyone be looking up the dose of oseltamivir for other reasons (ie teaching/learning?)

Response: This is an important point. We have added the following sentence in the Discussion section: “Some searches may have been performed by medical students for learning purposes or senior physicians in teaching situations. In addition, HCPs other than physicians (e.g., nurses, pharmacists) may have searched for information on oseltamivir. However, we assume that the majority of oseltamivir searches occur in practice in primary and specialized care by clinical physicians.”

Comment: Could people be prescribing without looking up the dose once they know it?

Response: This is possible. We have added the following sentence in Discussion: “It is also worth noting that some HCPs may be familiar with the correct oseltamivir dosage once verified, thus a database search is not performed, especially when a physician regularly encounters children with similar weights or ages.”

Comment: The abstract doesn't read properly, and could do with checking for basic English.

Response: We have proofread the manuscript throughout, especially the abstract. The text has been language-checked, edited, and proofread in terms of proper English language. Revisions in English language can be seen in the file labeled “Revised Manuscript with Tracked Changes”.

---

## [Editor Report · Decision Letter 2]

13 Jul 2022

Online searches of children’s oseltamivir in public primary and specialized care: Detecting influenza outbreaks in Finland using dedicated databases for health care professionals

PONE-D-21-11786R2

Dear Dr. Mukka,

We’re pleased to inform you that your manuscript has been judged scientifically suitable for publication and will be formally accepted for publication once it meets all outstanding technical requirements.

Kind regards,

Ahmad Khanijahani

Guest Editor

PLOS ONE
---

## [Editor Report · Acceptance letter]

27 Jul 2022

PONE-D-21-11786R2 

Online searches of children’s oseltamivir in public primary and specialized care: Detecting influenza outbreaks in Finland using dedicated databases for health care professionals 

Dear Dr. Mukka:

I'm pleased to inform you that your manuscript has been deemed suitable for publication in PLOS ONE. Congratulations! Your manuscript is now with our production department. 

Kind regards, 

on behalf of

Dr. Ahmad Khanijahani 

Guest Editor

PLOS ONE